# Information Survey on the Use of Complementary and Alternative Medicine

**DOI:** 10.3390/medicina58010125

**Published:** 2022-01-14

**Authors:** Marco Paoloni, Francesco Agostini, Sergio Bernasconi, Gianni Bona, Carlo Cisari, Massimo Fioranelli, Marco Invernizzi, Antonello Madeo, Marco Matucci-Cerinic, Alberto Migliore, Nicola Quirino, Carlo Ventura, Roberto Viganò, Andrea Bernetti

**Affiliations:** 1Department of Anatomical and Histological Sciences, Legal Medicine and Orthopedics, Sapienza University of Rome, 00185 Rome, Italy; francesco.agostini@uniroma1.it (F.A.); andrea.bernetti@uniroma1.it (A.B.); 2Italian Society for Pediatric Endocrinology and Diabetology (SIEDP), 43126 Parma, Italy; sbernasconi3@gmail.com; 3Department of Health Sciences, University of Piemonte Orientale, 28100 Novara, Italy; gianni.bona@maggioreosp.novara.it (G.B.); carlo.cisari@maggioreosp.novara.it (C.C.); marco.invernizzi@med.uniupo.it (M.I.); 4Department of Human Sciences, Guglielmo Marconi University, 00193 Rome, Italy; massimo.fioranelli@gmail.com; 5Translational Medicine, Dipartimento Attività Integrate Ricerca e Innovazione (DAIRI), Azienda Ospedaliera SS Antonio e Biagio e Cesare Arrigo, 15121 Alessandria, Italy; 6DISTU (Language, History, Philosophy and Law) Department, University of Tuscia, 01100 Viterbo, Italy; antonellomadeo@gmail.com; 7Department of Experimental and Clinical Medicine, Division of Rheumatology AOUC, University of Florence, 50121 Florence, Italy; cerinic@hotmail.com; 8Unit of Rheumatology, San Pietro Fatebenefratelli Hospital, 00189 Rome, Italy; reumafbf@libero.it; 9Business School, LUISS “Guido Carli” University, 00197 Rome, Italy; quirinonicola@gmail.com; 10Department of Experimental, Diagnostic and Specialty Medicine (DIMES), School of Medicine, University of Bologna, 40126 Bologna, Italy; ventura.vid@gmail.com; 11Azienda Socio-Sanitaria Territoriale Gaetano Pini-CTO, 20122 Milan, Italy; vigano2008@libero.it

**Keywords:** Delphi consensus, acupuncture, homeopathy, osteopathy, phytotherapy, Chinese medicine

## Abstract

*Background and Objectives*: Complementary and alternative medicines (CAMs) are generally considered non-scientific and poor effective therapies. Nevertheless, CAMs are extensively used in common clinical practice in Western countries. We decided to promote a Delphi consensus to intercept the opinion of Italian physicians on CAM use in clinical practice. *Materials and Methods*: We run a Delphi-based consensus, interviewing anonymously 97 physicians. Of these, only 78 participate to the questionnaire. *Results*: Consensus about agreement and disagreement have been reached in several topics, including indication, as well as safety issues concerning CAMs. *Conclusions*: The use of CAMs in clinical practice still lacks evidence. Experts agree about the possibility to safely use CAMs in combination with conventional medicines to treat non-critical medical conditions.

## 1. Introduction

The term complementary and alternative medicine (CAM) describes any practice that aims to achieve the healing effects of medicine, with uncertain plausibility, being untested and/or unverifiable using scientific methods [1,2,3]. For these reasons, they are not included in the field of scientific medicine [4].

It should be noted, however, that a given practice could be considered CAM in one country, and conventional medicine in another country. Hence, the Cochrane Collaboration states that “alternative medicine includes all those practices and ideas that arise outside the medical mainstream in many countries” [5].

The US National Center on Complementary and Integrative Health (NCCIH), has created a classification system for the branches of CAM that divides them into five main groups [6]: (I) whole medical systems (Chinese medicine, naturopathy, homeopathy, and Ayurveda); (II) mind–body interventions; (III) practices based on biology (i.e., traditional Chinese medicine and other non-biological substances); (IV) manipulative and body-based practices: (chiropractic and osteopathic manipulation); (V) energy medicine (biofield therapies and bioelectromagnetic-based therapies).

In Italy, according to the guidelines issued by the National Council of the Fnomceo (in Italian, Federazione Nazionale degli Ordini dei Medici Chirurghi e degli Odontoiatri—National Federation of Physicians, Surgeons, and Dentists) in 2002, only nine CAM are considered relevant from a social point of view, according to the indications of the European Parliament [7] and of the Council of Europe [8]: (I) acupuncture; (II) phytotherapy; (III) anthroposophical medicine; (IV) Ayurvedic medicine; (V) homeopathic medicine; (VI) traditional Chinese medicine; (VII) homotoxicology; (VIII) osteopathy; (IX) chiropractic. They are considered diagnostic, therapeutic, and preventive systems that work alongside official medicine [9]. Assuming that any therapeutic intervention should be preceded by a correct diagnosis, CAM should be a medical act of exclusive competence and professional responsibility of the doctor, dentist, veterinarian, and pharmacist, each for their respective competences. On 7 February 2013, an agreement was issued in the Italian Permanent State-Regions Conference to regulate the professional profile of doctors who practice acupuncture, phytotherapy, and homeopathy, establishing professional registers [9]. There is a general scientific consensus that CAMs lack the necessary scientific validation, and their effectiveness is not proven, or even denied [10,11]. Many of the claims about the efficacy of CAMs are controversial, as research on them is often of low quality and methodologically incorrect [12].

Nevertheless, CAMs are extensively used in common clinical practice in Western countries. In 1998, a systematic review concluded that about 13% of cancer patients used some form of CAM [13]. Recent data show that nowadays, one in three patients with a neuro-muscular disorder has already gone through an osteopath or a chiropractor, or is being treated with acupuncture, or taking homotoxicologic or homeopathic products not for ideological reasons, but for therapeutic necessity: complications and/or intolerances to traditional drugs; or dissatisfaction or ineffectiveness of the treatment undertaken with official medicine [14]. More generally, in Italy [15], over one citizen out of five (21.2% of the population) uses CAM (with an increase of 6.7% compared to 2012), and homeopathy is the alternative cure more widespread. In fact, when deciding not to rely on traditional medicine, they are moving towards homeopathy (76.1% of patients who decide to be treated with CAM), followed by phytotherapy (58.7%), osteopathy (44.8%), acupuncture (29.6%), and, finally, chiropractic (20.4%) [15]. Today, the number of patients who choose CAMs is over 12.8 million, whereas in 2000, they were just over 6 million for all CAMs. The number of doctors who prescribed CAMs in 2000 was about 5000, and today that number is more than 20,000 [16].

Considering the large number of physicians taking care of their patients with CAMs, to know how they manage patients and pathologies, as well as how they care about the safety of patients when choosing an alternative (and not evidence-based) approach is mandatory.

The aim of the present study has been, therefore, to gather, by using a Delphi method, the opinions of a group of Italian physicians, experts in the use of CAMs, who are involved in the management of various diseases. Particularly, our goal has been to identify the main aspects involved in patient selection, the choice of therapeutic agents, and the safety profile, as well as the legal and organizational aspects, to obtain opinion-based recommendations to be used in daily clinical practice.

## 2. Materials and Methods

The Delphi method has been used to conduct this consensus initiative [17,18]. A committee of 13 experts from Italian universities, public hospitals, territorial services, and research institutes formed the Consensus Board. Physicians with experience in the use of complementary and alternative medicines, ascertained by scientific publications on the subject, were considered. Physicians who had declared a possible conflict of interest were excluded. The Consensus Board reviewed the literature from November 2019 to February 2020, and, on the basis of the CAMs currently identified in Italy by Fnomceo (acupuncture; phytotherapy; anthroposophical medicine; Ayurvedic medicine; homeopathic medicine; traditional Chinese medicine; homotoxicology; osteopathy; chiropractic), developed the first-round questionnaire (Q1) [19]. Q1 has been submitted to a group of CAM experts, with different specializations, who were selected from among the largest Italian medical centers, specialized in CAM therapy, by means of a non-probability sampling method. The experts received an email in which the rationale and the aims of the research were explained. The email sent to each physician contained a strictly personal link to Q1 that allowed the questionnaire to be filled in online. Three reminder emails were sent to each expert in the 30-day period within which the Q1 had to be returned. To determine the consensus level, the answers to each question have been grouped into three tertiles according to the Likert scale scores (1–3: disagreement; 4–6: neutral; 7–9: agreement).

All the items in which consensus was weak (i.e., 50–65% of answers) were included again in a second-round questionnaire (Q2) to those physicians who completed Q1. The results yielded by Q2 were analyzed in the same way as those yielded by Q1.

The final analysis was followed by a board meeting held to discuss the results, and to announce the final recommendations by the Consensus Board.

## 3. Results

A total of 97 doctors, with different specializations, were invited to attend the consensus conference. Of these, only 78 (76%) sent the correctly completed questionnaire, and constituted our sample. Several medical specialties have been represented in our sample; particularly, we had physicians specialists in physical and rehabilitation medicine (4.68%), cardiovascular diseases (0.78%), nutrition and food science (0.78%), gynecology and obstetrics (5.46%), general surgery (0.78%), dentistry and odontostomatology (11.7%), clinical biochemistry (0.78%), anesthesiology, pain and intensive care (2.34%), pediatrics (7.02%), general pathology (0.78%), internal medicine (1.56%), toxicology (0.78%), sports medicine (0.78%), orthopedics and traumatology (0.78%), radiology (0.78%), diseases of the respiratory system (0.78%), thermal medicine (0.78%), occupational medicine (0.78%), nephrology (0.78%), psychiatry (1.56%), endocrinology and metabolism (0.78%), plastic, reconstructive and aesthetic surgery (1.56%), vascular surgery (0.78%), pathological anatomy (0.78%), otorhinolaryngology (0.78%), angiology (0.78%), general practitioners (11.7%), and physicians without specialization (10.14%). Three doctors have a double specialization: occupational medicine and general surgery; cardiology, and physical and rehabilitative medicine; internal medicine and cardiology.

More than half (65.4%) of responders answered that they use CAM in more than half of their patients, the most of them being experts in homotoxicology (97.4%), phytotherapy (52.6%), homeopathy (43.6%), acupuncture (39.7%), traditional Chinese medicine (21.8%), and osteopathy (11.5%). In regards to the age of patients, the doctors in our sample reserve CAM treatments to patients in all the age ranges. Particularly, the age ranges more represented are those of 31–50 years (88.5%), 51–70 years (82.1%), and 16–30 years (64.1%). The most important reason to choose CAM as therapy is a previous failure or ineffectiveness of traditional medicine, followed by the occurrence of collateral/adverse events with traditional medicine, comorbidity that contraindicates the use of conventional medicine, patient’s personal choice, and physician’s personal choice.

In regards to the Delphi section, by analyzing data from both questionnaires, the Board identified statements that attained a consensus more than 75%, which led to the definition of the recommendations both for agreement and for disagreement (Appendix A).

## 4. Discussion

The purpose of the present research has been to identify items defining the common clinical attitudes of a group of Italian doctors, experts in CAM, regarding its application in daily practices. It is well known, in fact, that CAM is often viewed as a sort of “magical” or “mysterious” medical practice. The common view of CAM practices as not evidence-based is often the expression of such a believing attitude. However, some of these therapies have proven their efficacy according to evidence-based medicine principles. Our survey moves in the direction to understand how Italian doctors’ experts in CAM approach their use and prescription.

According to our results, doctors in our sample agree that CAM represents a useful integration to conventional medical practices. Apart from osteopathy and chiropractic, CAM should be practiced by a medical doctor licensed to practice medicine and, in any case, the use of CAM should always be preceded by a medical diagnosis. From this point of view, it should be noted that, differently from US [20], in Italy and EU, osteopathy and chiropractic are practiced by health professionals different from medical doctors (physiotherapists), with education often considered as unregulated [21], and this can explain the differentiation between these specialties and the others represented in the survey.

Interestingly, CAMs are considered by doctors in our sample as an integration to conventional therapies. Our experts, in fact, agreed that CAM and conventional medicine can be used contemporarily on the same patient, if needed. For the same reason, no contraindications have been identified in the use of CAM together with vaccinations. This is in line with recent findings that highlighted how, despite vaccine hesitation seeming to be more pronounced among CAM users, it is more probably due to distrust in conventional medicine rather than in trust in CAM [22]. Creating adequate knowledge, and reducing hesitation about the use of vaccines could therefore be an important goal for those doctors who practice CAM as well. Several clinical conditions can be treated by CAM according to our experts.

Particularly, musculoskeletal conditions benefit from integration between conventional medicine and the following CAMs: acupuncture, homeopathy, homotoxicology, osteopathy, traditional Chinese medicine, and phytotherapy. Moreover, more than one CAM can be used in combination on the same patient, if needed. Musculoskeletal pathologies are, without a doubt, among the most important indications for CAM practitioners. Despite low evidence of efficacy coming from clinical studies and systematic reviews [23,24,25,26,27], it is generally accepted that some approaches may improve pain and function in chronic conditions, such as osteoarthritis [28].

Dermatologic pathologies may be treated by means of an integrated approach of conventional and CAMs, particularly homeopathy, homotoxicology, phytotherapy, and traditional Chinese medicine. CAMs are often used by patients suffering from dermatologic conditions, especially atopic dermatitis, acne, and psoriasis. Unfortunately, even in this case, the large number of users cannot be completely supported by proper data coming from research. It seems, in fact, that data from clinical trials are still insufficient to allow the use of CAM according to evidence-based medicine principles [29], thereby not being recommended in clinical settings [30].

No consensus about the use of acupuncture on dermatologic disease has been achieved in this survey. However, recent data show that acupuncture improves clinical outcomes in some dermatologic pathologies, including uremic pruritus, atopic dermatitis, urticaria, and itch, despite the remarked need for additional large-scale, randomized, sham-controlled trials to confirm the role of acupuncture in dermatology [31].

Respiratory system pathologies benefit from the integration between conventional and CAMs, particularly homeopathy, homotoxicology, phytotherapy, acupuncture, and traditional Chinese medicine. A recent systematic review shows that Chinese oral herbal therapies are effective in reducing exacerbations in chronic obstructive pulmonary diseases [32], and different cases of asthma, in which current evidence does not support the systematic use of CAM in adults and children [33].

Allergic/immunological pathologies benefit from the integration between conventional and CAMs, particularly homeopathy, homotoxicology, acupuncture, and traditional Chinese medicine.

Endocrine pathologies benefit from the integration between conventional medicine and CAMs, particularly homeopathy, homotoxicology, and phytotherapy. However, it should be noted that research data are not encouraging the use of CAM in several endocrine pathologies, including polycystic ovarian syndrome [34] or diabetic kidney disease [35]. Physicians must be aware, therefore, that the use of CAM is not supported in diseases requiring specific medical treatments [36]. In this case, however, it should be noted that the experts of our survey stated that CAMs should not be used alone when a replacement therapy (e.g., insulin or hormone replacement therapy) is needed.

Headache and neck pain benefit from the integration between conventional and CAMs, particularly homeopathy, homotoxicology, acupuncture, chiropractic, osteopathy, and traditional Chinese medicine.

Cardiovascular diseases benefit from the integration between conventional and CAMs, particularly homeopathy and homotoxicology. Despite not being considered as a therapeutic approach for cardiovascular diseases, it can be observed that, in some cases, CAM have been considered as adjunctive therapies, for example, to control some risk factors for cardiovascular diseases, with poor results [37]. Also, some techniques not included in this survey, such as, for example, all those related to mindfulness, might be useful to control stress factors, considered important in the genesis of cardiovascular disease.

Gastrointestinal disorders benefit from the integration between conventional and CAMs, particularly homeopathy, homotoxicology, acupuncture, and traditional Chinese medicine.

Nephro-urology diseases benefit from the integration between conventional and CAMs, particularly homeopathy, homotoxicology, and traditional Chinese medicine.

Neuropsychiatric disorders benefit from the integration between conventional and CAMs, particularly homeopathy and homotoxicology.

In regards to oncology, in line with similar experiences reported in the literature [38,39,40], experts in our survey believe that CAM is useful to reduce collateral and adverse events during cancer chemo- and/or radiotherapy, as well as to give psychological support to the patients. CAMs are not considered as a therapy for the modification of the natural history of the pathology nor to prevent metastatic diffusion.

One of the most important applications of CAM can be surely found in pain treatment. We tried to understand if different pain diagnoses could differently respond to CAM, according to our experts. According to our results, acupuncture, homotoxicology, and osteopathy can be recommended in combination with conventional medicine to treat nociceptive, neuropathic, and muscle-skeletal pain. In this latter condition, patients can also benefit from the use of homeopathy. Worthy of note, experts in our sample agreed that, in the case of a patient exclusively treated with CAM that does not respond to therapies and/or progressively gets worse, doctors should stop CAM treatment, and shift to appropriate conventional therapies as needed. CAMs are usually considered safe. According to our experts, complications derived from CAM are not frequent if homeopathy, homotoxicology, or acupuncture are used.

The main limitation of this article is that the present results reflect the opinion of a sample of physicians. Despite trying to select a sample of experts in the field of CAM, we found, according to experts’ answers, a low-level of expertise in some specialties, particularly chiropractic, Ayurvedic medicine, and osteopathy. This could reflect the limited agreement on these specific CAM sub-specialties.

## 5. Conclusions

The use of CAMs in clinical practice still lacks evidence. Experts agree about the possibility to safely use CAM in combination with conventional medicines to treat non-critical medical conditions. In conclusion, this article has highlighted several consensus statements about CAM indications and contra-indications, which could be useful to project future trials, needed to assess the efficacy and effectiveness of CAM, as well as its safety.

## Data Availability

Data available on request. The data presented in this study are available on request from the corresponding author.

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
