# Peer review of "Information Survey on the Use of Complementary and Alternative Medicine"

_medicina, 2022, doi:10.3390/medicina58010125_

Round 1

Reviewer 1 Report

Dear  authors,

Please see the attached file comments and revised the manuscript point by point.

Truly yours,

Journals referee

Author Response

Dear Reviewer 1, thank you for your comment. We have edited the text following your suggestions. Regarding the realization of the questionnaire, we referred to the literature on the topic that we also added as a reference to our manuscript.

- McGlynn EA, Kosecoff J, Brook RH. Format and conduct of consensus development conferences. Multi-nation comparison. Int J Technol Assess Health Care. 1990; 6 (3): 450-69. doi: 10.1017 / s0266462300001045. PMID: 2228459

Reviewer 2 Report

 The major aim of the presented article is to gather "the opinions of a group of Italian physicians, experts in the use of CAMs, who are involved 89 in the management of various diseases" (p 2). As the headline describes it is an " Information survey on the use of complementary and alternative medicine". In the Discussion the purpose is defined as "to identify items defining the common 170 clinical attitude of a group of Italian doctors, experts in CAM, regarding its application in 171 daily practice (p 7)".

For this purpose, a survey questionnaire is sufficient, and still enable to statistically analyze the answers. No Delphi procedure is required. The use of the Delphi Method in the present article is only for the formation and consolidation of the survey questionnaire, and seems to be of minor importance regarding the aims and purposes of the study.

The Delphi is a consensus process which addresses areas of debate or uncertainty through expert consensus.  The consensus is reached through several rounds of discussions, usually anonymous. But here there was no debate or uncertainty.

So it seems that the Delphi process is too bold and salient in the article, and should be more minor.

I recommend rewriting the article, changing the title to " Information survey on the use of complementary and alternative medicine" only, reducing the part of the Delphi in the article to minimum, and moving the tables to the appendix. Thus the emphasis will be on the results rather than on the process, which is of secondary importance here.  

Author Response

Reviewer 2: The major aim of the presented article is to gather "the opinions of a group of Italian physicians, experts in the use of CAMs, who are involved 89 in the management of various diseases" (p 2). As the headline describes it is an " Information survey on the use of complementary and alternative medicine". In the Discussion the purpose is defined as "to identify items defining the common 170 clinical attitude of a group of Italian doctors, experts in CAM, regarding its application in 171 daily practice (p 7)". For this purpose, a survey questionnaire is sufficient, and still enable to statistically analyze the answers. No Delphi procedure is required. The use of the Delphi Method in the present article is only for the formation and consolidation of the survey questionnaire, and seems to be of minor importance regarding the aims and purposes of the study. The Delphi is a consensus process which addresses areas of debate or uncertainty through expert consensus.  The consensus is reached through several rounds of discussions, usually anonymous. But here there was no debate or uncertainty. So it seems that the Delphi process is too bold and salient in the article, and should be more minor. I recommend rewriting the article, changing the title to " Information survey on the use of complementary and alternative medicine" only, reducing the part of the Delphi in the article to minimum, and moving the tables to the appendix. Thus the emphasis will be on the results rather than on the process, which is of secondary importance here.  

Authors: Dear Reviewer, thank you for your comment. We have changed the title and text of our manuscript as you suggested, also moving the tables as supplementary material.